# Observation of frustrated chiral dynamics in an interacting triangular flux ladder

Yuqing Li [1,2], Huiying Du[1], Yunfei Wang[1], Junjun Liang[1], Liantuan Xiao[1,2], Wei Yi [3,4,5] ✉, Jie Ma [1,2,5] ✉ & Suotang Jia [1,2]

Quantum matter interacting with gauge fields, an outstanding paradigm in modern physics, underlies the description of various physical systems. Engineering artificial gauge fields in ultracold atoms offers a highly controllable access to the exotic many-body phenomena in these systems, and has stimulated intense interest. Here we implement a triangular flux ladder in the momentum space of ultracold $^{133}$Cs atoms, and study the chiral dynamics under tunable interactions. Through measurements of the site-resolved density evolutions, we reveal how the competition between interaction and flux in the frustrated triangular geometry gives rise to flux-dependent localization and biased chiral dynamics. For the latter in particular, the symmetry between the two legs is dynamically broken, which can be attributed to frustration. We then characterize typical dynamic patterns using complementary observables. Our work opens the avenue toward exploring correlated transport in frustrated geometries, where the interplay between interactions and gauge fields plays a key role.

Understanding the interplay of interactions and gauge fields is an outstanding task in contemporary physics. From the emergence of fundamental particles to the fractional quantum Hall effects, such an interplay proves essential in the description of physical phenomena with ranging energy and temperature scales[1–4]. The implementation of synthetic gauge fields in ultracold atoms offers a flexible playground where both the interaction and gauge field are highly tunable in a quantum many-body environment[5–9]. This opens up rich possibilities for exploring, for instance, interaction-induced chiral dynamics[10], and the quantum simulation of strongly interacting topological systems[11,12]. While cold atoms are charge neutral, different forms of synthetic gauge fields can be engineered through rapidly rotating atomic gases[13,14], Raman transitions[15,16], lattice shaking[17,18], or laser-induced hopping[19–26].

The recently developed momentum-lattice technique further facilitates the quantum simulation of gauge fields with ultracold atoms[27,28]. Therein, the synthetic lattice dimension is encoded in the

atomic momentum states, and the lattice hoppings are generated by parametric laser couplings whose amplitudes and phases are tunable in a site-resolved manner[29–33]. Moreover, momentum lattices also provide a direct access to exotic lattice geometry and hopping pattern. For instance, both square and triangular flux ladders were experimentally implemented, and the flux-dependent atomic transport and chiral dynamics were observed with single-site resolution[34–36]. These experiments, however, are mostly restricted to the non-interacting or weakly interacting regime, leaving the interplay between interaction and synthetic flux largely unexplored. Such an interplay is particularly intriguing in lattices with frustrated geometry, where nontrivial ground states and correlated transport are expected to emerge[37,38].

In this work, we experimentally characterize the transport dynamics in an interacting triangular flux ladder, synthesized by adopting the momentum-lattice technique in a Bose–Einstein condensate (BEC) of $^{133}$Cs atoms. A broad Feshbach resonance,

[1]State Key Laboratory of Quantum Optics and Quantum Optics Devices, Institute of Laser Spectroscopy, College of Physics and Electronics Engineering, Shanxi University, Taiyuan 030006, China. [2]Collaborative Innovation Center of Extreme Optics, Shanxi University, Taiyuan 030006, China. [3]CAS Key Laboratory of Quantum Information, University of Science and Technology of China, Hefei 230026, China. [4]CAS Center For Excellence in Quantum Information and Quantum Physics, Hefei 230026, China. [5]Hefei National Laboratory, Hefei 230088, China. ✉e-mail: wyiz@ustc.edu.cn; mj@sxu.edu.cn

combined with the flexibility afforded by the momentum lattice, enables unprecedented access to the interplay of interaction, flux, and triangular-lattice geometry. By measuring the site-resolved atom-density evolutions, we study the impact of interactions on the flux-induced chiral dynamics of atoms initialized on a single lattice site. We show that the chiral transport is suppressed under strong interactions, where the atoms become localized to the initial site. Under intermediate interaction and flux, however, the atoms exhibit biased chiral dynamics, where the chiral transport persists but atoms predominately occupy one of the two legs, dynamically breaking the symmetry of the ladder. The biased chiral dynamics, absent in a square flux ladder, can be attributed to the frustrated triangular geometry. We reveal the interaction- and flux-dependent transitions between regimes with balanced and biased chiral dynamics, and with interaction-induced localization.

## Results

### Implementing the triangular flux ladder
Our experiment starts with a BEC of $4 \times 10^{4}$ $^{133}$Cs atoms in the hyperfine state $|F=3, m_F=3\rangle$ in a cigar-shaped optical trap[39]. A pair of counter-propagating laser beams with wavelength $\lambda = 1064$ nm are used to illuminate the weakly trapped BEC (see Fig. 1a) and drive a series of two-photon Bragg transitions between discrete atomic momentum states $p = 2m\hbar k$, where $\hbar$ is the reduced Planck's constant, $k = 2\pi/\lambda$, and $m$ labels discrete momentum states, taking integer values in the range [−6, 7]. The nearest-neighbor (NN) momentum states with an increment of $2\hbar k$ are thus coupled, forming the one-dimensional (1D) momentum lattice illustrated in Fig. 1b[27,28,40]. In addition, the four-

photon Bragg transitions are introduced to couple the next-nearest-neighbor (NNN) momentum states (see Supplementary Note 1)[35,36]. A triangular ladder consisting of 2 × 7 sites is subsequently visualized by mapping atomic momentum states into sites along different legs of the ladder, where the site index $(\ell, j)$ indicates the $\ell$th site along the $j$th leg (see Fig. 1c). The mapping relation between the discrete momentum states and sites in the synthetic triangular ladder is given by $m = 2\ell + j$ (see Supplementary Note 2 and Supplementary Fig. 1). Here $\ell$ takes integer values in the range [−3, 3], and $j \in \{0, 1\}$. With precise control of the NN and NNN coupling strengths, we individually set the inter- and intra-leg hopping rates as $J/h = 0.4$ kHz and $K/h = 0.2$ kHz, respectively. By imposing phases with alternating signs on the NNN couplings (see Fig. 1b), we thread each triangular plaquette with a synthetic flux $\phi$.

### Theoretical model and chiral dynamics
In the absence of interactions, the dynamics of the atoms in the synthetic ladder is governed by the effective Hamiltonian (see Supplementary Note 3)

$$
\begin{aligned}
H_0 = \quad & -J \sum_{\ell} (\hat{c}_{\ell,1}^{\dagger} \hat{c}_{\ell,0} + \hat{c}_{\ell,1}^{\dagger} \hat{c}_{\ell+1,0}) \\
& -K \sum_{\ell,j} e^{i(-1)^j \phi} \hat{c}_{\ell+1,j}^{\dagger} \hat{c}_{\ell,j} + \text{h.c.},
\end{aligned} \tag{1}
$$

where $\hat{c}_{\ell,j}^{\dagger}$ and $\hat{c}_{\ell,j}$ are the creation and annihilation operators for atoms with site label $(\ell, j)$. The short-range inter-atomic interactions, on the other hand, lead to long-range interactions in the momentum space (see Fig. 1c). To provide insights into the interaction effects, we follow

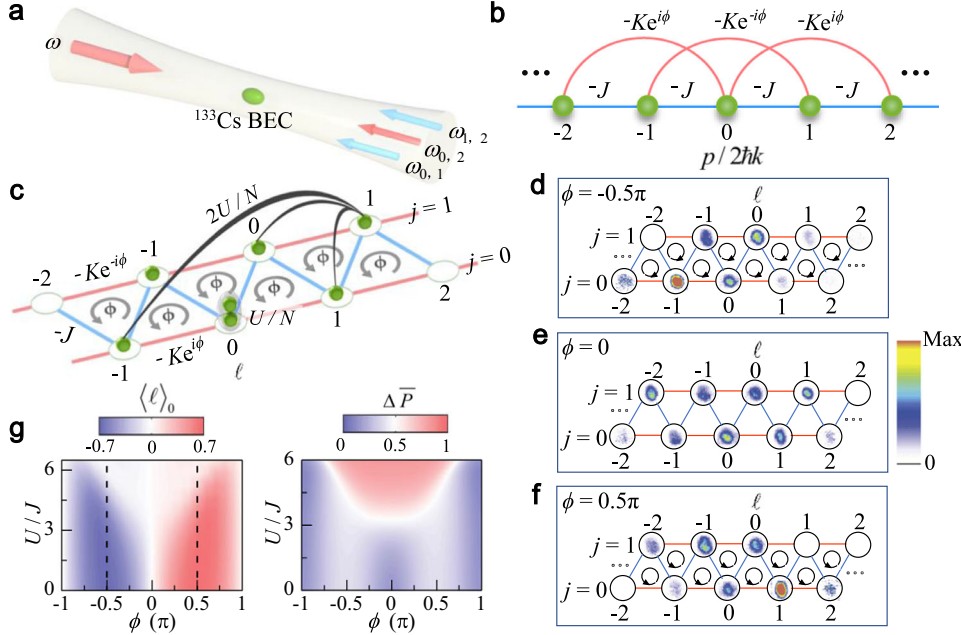

**Fig. 1 | Synthesizing triangular ladders with tunable flux and interaction. a** A $^{133}$Cs BEC is illuminated by two counterpropagating, far-detuned laser beams, one of which contains multifrequency components. **b** These laser fields drive a series of two-photon Bragg transitions to a couple of nearest-neighbor (NN) momentum states $p = 2m\hbar k$ ($m = -6,...,7$) for synthesizing a 1D momentum lattice. A zigzag ladder with site index $(\ell, j)$ is formed by adding next-nearest-neighbor (NNN) hoppings created by four-photon Bragg transitions. **c** The inter- and intra-leg tunneling rates are characterized by $J$ and $K$, respectively. The applied tunneling phases in the NNN hoppings create flux $\phi$ in each triangular plaquette. Atoms in the same site experience an interaction strength $U/N$ (gray shadow) with the mean-field interaction energy $U$, and atoms in different sites have an interaction strength $2U/N$.

(black lines). **d**−**f** The experimentally observed population distributions of atoms in the ladder in the noninteracting regime. The population distribution is constructed by rearranging 1D optical density images taken after the evolution time $t = 600$ μs and 22-ms time of flight for fluxes $\phi = 0, \pm 0.5\pi$. **g** Typical dynamic patterns under the interplay between interaction and flux. Left: the variation of average site index $\langle \ell \rangle_0$ in the $j = 0$ leg at the evolution time $t = 1.5 \, \hbar/J$ with the flux $\phi$ and interaction $U/J$. Right: the average inter-leg polarization $\Delta \overline{P}$ calculated within the evolution time $t = 2 \, \hbar/J$ as a function $\phi$ and $U/J$. For all experiments and numerical simulations, we initialize the atoms on the site $(\ell = 0, j = 0)$, corresponding to the ground state of a non-interacting BEC.

the widely adopted mean-field approximation in momentum lattices[31,32,35,40,41], and write down the effective interaction Hamiltonian (see Methods)

$$H_{\text{int}} = U\left(N - \frac{1}{2}\right) - \frac{U}{2N}\sum_{\ell,j} n_{\ell,j}^2,$$ (2)

where $N$ is the total atom number and $U$ is proportional to the atomic $s$-wave scattering length $a$[42,43]. For $a > 0$, the atomic interactions give rise to a density-dependent, local attractive potential, leading to nonlinear self-trapping when the interaction is strong enough[31,41,44,45]. In previous studies with the momentum lattice, the interaction energy is adjusted through the atomic density or tunneling energy[31,32]. For $^{133}$Cs atoms, the scattering length is tunable through a broad Feshbach resonance[7], enabling a convenient control of the interaction term.

For numerical simulations below, we adopt a mean-field approach, setting $\langle c_{\ell,j}\rangle = \tilde{\varphi}_{\ell,j}$ in their Heisenberg equations of motion, and evolve the resulting coupled Gross-Pitaevskii equations to get the time-dependent condensate wavefunctions $\tilde{\varphi}_{\ell,j}$ (see Supplementary Note 3). It is important to note that, while our mean-field approach offers a transparent and qualitatively valid understanding of the interaction effects (see discussions below), the experimental results are necessarily beyond-mean-field.

For non-interacting or weakly interacting flux ladders, a typical dynamic feature is the chiral transport, where atoms initialized in a single site flow in different directions on different legs. The chiral dynamics can be probed experimentally by measuring the density evolution of atoms initialized in the site (0, 0). After an evolution time, the atomic density distribution on the ladder can be constructed by mapping the momentum-space 1D optical density (OD) distribution, which is obtained by a 22-ms time-of-flight (TOF) image, onto that over individual sites in the ladder. In Fig. 1d–f, we show the site-resolved density distribution in the absence of interaction following an evolution time $t = 600$ μs. For positive (negative) flux $\phi$, atoms propagate to the right (left) on the $j = 0$ leg where they are initialized, while atoms in the other leg propagate in the opposite direction.

To quantitatively characterize the impact of interaction and flux on chiral dynamics, we define the average site index

$$\langle\ell\rangle_j = \sum_\ell \ell \times n_{\ell,j}(t),$$ (3)

which measures the displacement of the condensate along the $j$th leg. We further define the average inter-leg polarization to characterize dynamics of atoms along the rung direction

$$\Delta\overline{P} = \overline{P}_0 - \overline{P}_1,$$ (4)

where the time-averaged normalized population in the $j$th leg within an evolution time $t$ is given by

$$\overline{P}_j = \frac{1}{t}\int_0^t \sum_\ell n_{\ell,j}(t')dt'.$$ (5)

In Fig. 1g, we show the numerically simulated $\langle\ell\rangle_0$ (left) and $\Delta\overline{P}$ (right) as functions of the flux $\phi$ and interaction strength $U/J$. Combining the two complementary dynamic observables, three distinct dynamic regimes can be identified. Near $\phi = 0, \pm\pi$, and before the interaction becomes too strong, we have $|\langle\ell\rangle_0| > 0$ and $\Delta\overline{P} \sim 0$. Therein, the atoms exhibit balanced chiral dynamics, with almost equal time-averaged populations on the two legs. For intermediate $\phi$, the system exhibits biased chiral dynamics, where the chiral behavior persists with $|\langle\ell\rangle_0| > 0$, but $\Delta\overline{P}$ becomes appreciable. The atoms selectively occupy one of the legs, breaking the ladder symmetry (more discussion in Supplementary Note 5). For strong enough interactions, the atoms

become localized in the initial site with $\langle\ell\rangle_0 \sim 0$ and $\Delta\overline{P} \sim 1$, and the localization boundary is strongly dependent on the flux. Importantly, the biased chiral regime is absent in a square flux ladder (see Supplementary Note 6). Nor does the balanced chiral regime persist under strong interactions in a square ladder near $\phi = \pm\pi$, unlike the case of Fig. 1g, where $\phi = \pm\pi$ corresponds to the fully frustrated region of the triangular ladder. The emergence of these rich dynamic regimes in a triangular ladder are therefore closely related to the interplay of interaction, flux, and frustrated geometry.

## Characterizing dynamic behaviors

We now experimentally confirm the theoretical predictions above by measuring the chiral dynamics of atoms in the triangular ladder with tunable interaction $U/J$ and flux $\phi$. In Fig. 2b–e, we show the variations of atomic density distribution in the $j = 0$ leg (shaded in Fig. 2a) at the time $t = 1.5\hbar/J$ with increasing $U/J$ for different $\phi$. Chiral dynamics is clearly observed in the non-interacting and weakly interacting regimes (see also Supplementary Note 4). The atom transport is symmetric with respect to the initial site for $\phi = 0$, whereas atoms propagate in opposite directions for negative and positive fluxes. When the interaction becomes strong enough, the atoms become localized in the initial site. The flux-dependent localization is shown in Fig. 2f–h, where the average site indices $\langle\ell\rangle_0$ are extracted from the experimental data in Fig. 2b–e. With increasing $|\phi|$, the transition point moves to larger $U/J$, revealing the competition between interaction and flux.

We then measure the polarization dynamics characterizing the differential atomic population between the two legs (shaded in Fig. 3a). In Fig. 3b, we plot the average inter-leg polarization $\Delta\overline{P}$ as a function of $\phi$ under different $U/J$ for an evolution time $t = 2\hbar/J$. In the absence of interaction, $\Delta\overline{P}$ already deviates from zero for intermediate flux $|\phi|$ in the range $(0, \pi)$. Combined with the data in Fig. 2, this indicates biased chiral dynamics. For larger $U/J$, the atoms are mostly localized in the initial site with a large $\Delta\overline{P}$. Remarkably, under all experimentally implemented interactions, the measured $\Delta\overline{P}$ decreases toward zero in the vicinity of $\phi = \pm\pi$, which is consistent with the numerical prediction that the balanced chiral dynamics persists near $\phi = \pm\pi$. In Fig. 3c–e, we show the variations of the measured average atomic populations $\overline{P}_0$ and $\overline{P}_1$ with increasing $U/J$ for different $\phi$, from which we extract $\Delta\overline{P}$ as a function of $U/J$ for different $\phi$ in Fig. 3f–h. The inter-leg localization emerges at stronger interactions when the flux becomes larger, indicating the competition between interaction and flux.

Summarizing the data in Figs. 2 and 3, we map out regimes with distinct dynamic patterns. As shown in Fig. 4a, c, the variations of both the average site index $\langle\ell\rangle_0$ and the average inter-leg polarization $\Delta\overline{P}$ with $U/J$ and $\phi$ show excellent agreement with numerical simulations without any free parameters (see Fig. 4b, d). The measured parameter regimes with distinct dynamic behaviors are also consistent with the theoretical predictions in Fig. 1g.

## Discussion

We report the experimental study of frustrated chiral dynamics in an interacting triangular flux ladder. The experiment is facilitated by the momentum-lattice technique, in combination with the tunable interaction in the $^{133}$Cs condensate. Under the interplay of interaction, flux, and frustrated geometry, the system exhibits rich dynamic behaviors, consisting of regimes with balanced or biased chiral dynamics, and interaction-induced localization. We map out regimes with distinct dynamic patterns by measuring the density evolution and confirming the flux and interaction dependence of the dynamic transitions. Our work paves the way for understanding and engineering correlated transport in quasi-one-dimensional systems. For future studies, it is interesting to explore the impact of interaction on quantum phases and dynamics in synthetic flux lattices with inhomogeneous gauge fields or more exotic geometries. Here the inhomogeneous gauge

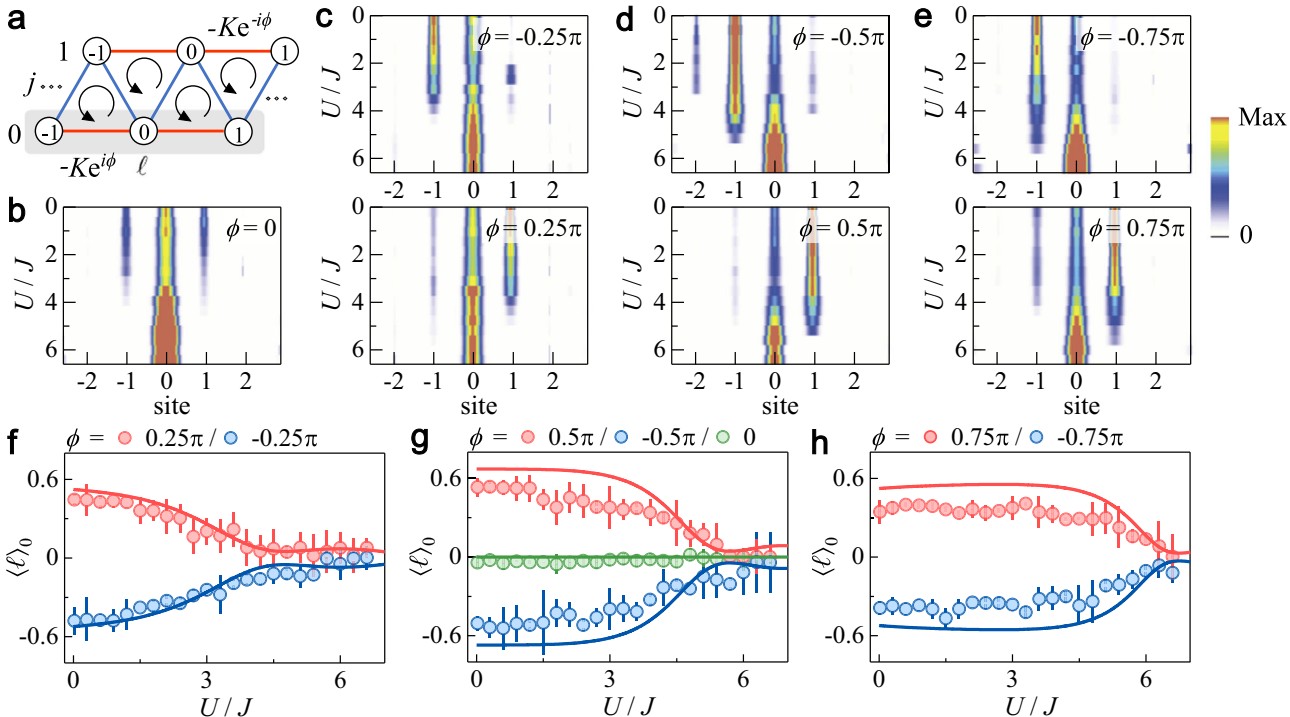

**Fig. 2 | Observation of chiral transport and flux-dependent localization.**
**a** Population distribution and average site index are measured in one leg (shaded) following an evolution time $t = 1.5\hbar/J$. **b**–**e** Measured population distribution in the $j = 0$ leg with increasing strength of interaction $U/J$ for different fluxes $\phi$.

**f**–**h** Average site index $\langle \ell \rangle_0$ extracted from the data in (**b**–**e**) as a function of $U/J$ for different $\phi$. The solid lines are the numerical simulations. All error bars denote standard errors. In all panels, the inter- and intra-leg hopping rates are $J/h = 0.4$ kHz and $K/h = 0.2$ kHz, respectively.

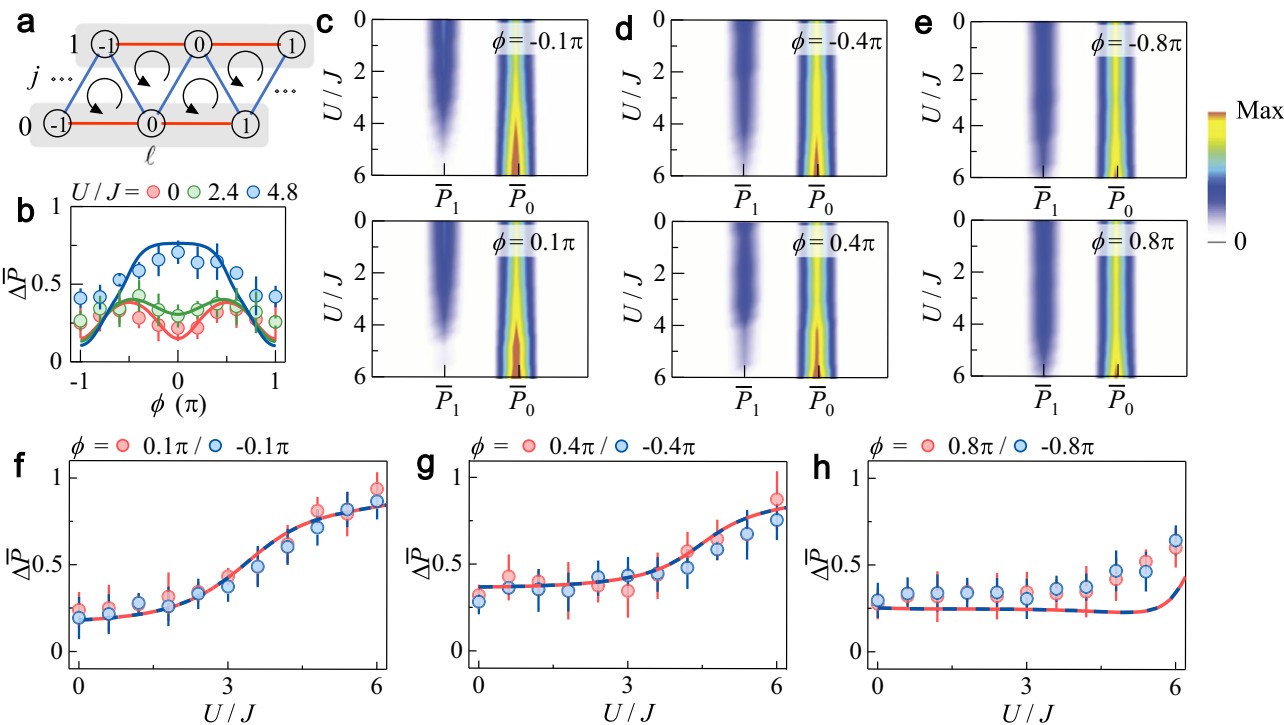

**Fig. 3 | Observation of the flux-dependent inter-leg polarization. a** Chiral dynamics of the atoms along the ladder is measured within an evolution time $t = 2\hbar/J$ for obtaining the averaged normalized population $\overline{P}_0$ ($\overline{P}_1$) in the $j = 0$ (1) leg. **b** Variation of the experimentally measured average inter-leg polarization calculated as $\Delta \overline{P} = \overline{P}_0 - \overline{P}_1$ with the flux $\phi$ for different interactions $U/J$. **c**–**e** Variations of

the measured average populations $\overline{P}_0$ and $\overline{P}_1$ with increasing $U/J$ for different $\phi$. **f**–**h** Average inter-leg polarization $\Delta \overline{P}$ extracted from the data in (**c**–**e**) as a function of $U/J$ for different $\phi$. The dashed lines are the numerical simulations. All error bars denote standard errors. In all panels, the inter- and intra-leg hopping rates are $J/h = 0.4$ kHz and $K/h = 0.2$ kHz, respectively.

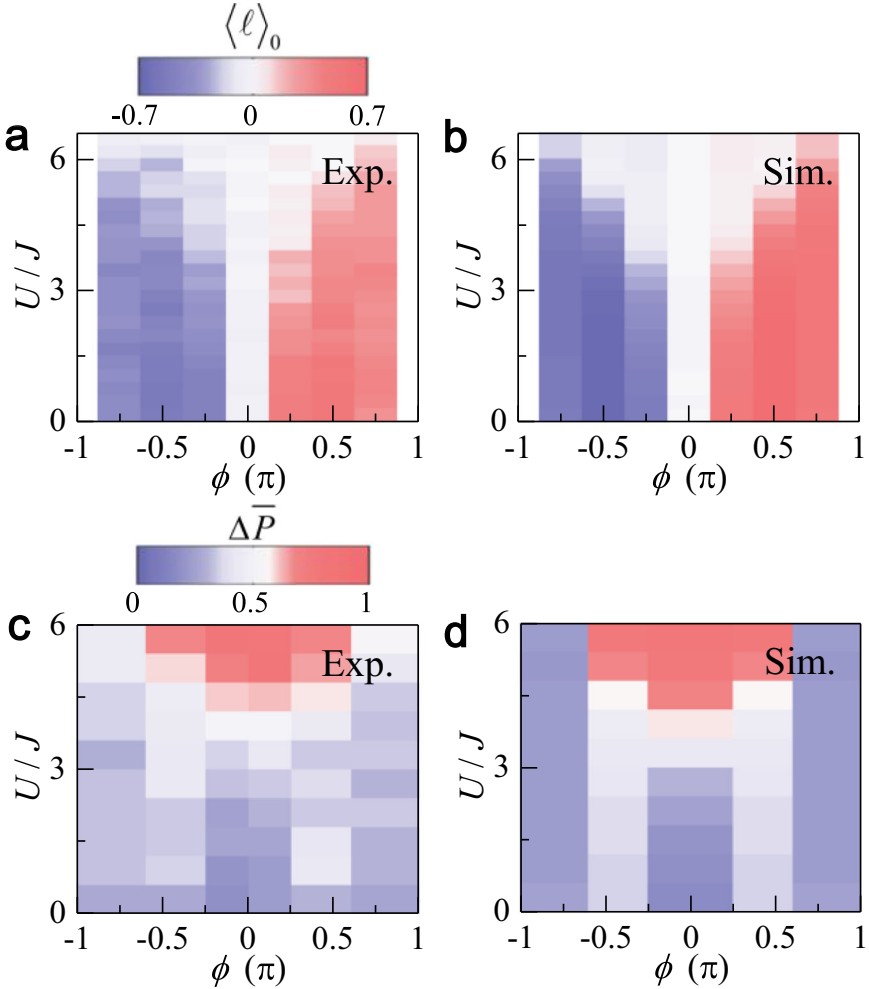

**Fig. 4 | Parameter regimes with distinct dynamic behaviors. a** Color contour of the measured average site index $\langle \ell \rangle_0$ as a function of interaction $U/J$ and flux $\phi$, and the corresponding numerical simulation shown in (**b**). **c** Color contour of the measured average inter-leg polarization $\Delta \overline{P}$ as a function of $U/J$ and $\phi$, and the corresponding numerical simulation shown in (**d**). All parameters are the same as those in Figs. 2 and 3.

fields can be implemented by individually controlling the tunneling phase of each laser-assisted hopping process along the legs[34]. More complicated lattice geometries can be achieved by parametrically coupling the desired momentum states with multi-photon processes, or by involving atomic hyperfine states in the lattice design[33, 36]. Last but not least, by introducing widely tunable interactions to a frustrated geometry, our experiment further calls for the development of beyond-mean-field descriptions which would be indispensable for engineering and understanding strongly correlated dynamics in the state-of-the-art momentum lattices.

## Methods
### Generating synthetic flux
The laser beam, which provides the strong radial confinement for $^{133}$Cs BEC in a cigar-shaped optical trap, is retro-reflected to illuminate the optically trapped atoms. We then use two acoustic optical modulators to imprint the multiple sidebands $\omega_{m,m+1}$ and $\omega_{m,m+2}$ on the reflected laser beam. These laser fields are used for generating a 1D momentum lattice with the additional NNN hoppings (see Fig. 1b). By rearranging the synthetic sites in the 1D momentum lattice, we create a triangular ladder, and the distinct momenta of the atoms in different sites enable the direct observation of atomic population on the site-resolved level.

By applying controllable phases in each of the multi-frequency components $\omega_{m,m+2}$, we impose alternating phases with opposite signs ($\pm \phi$) on the NNN hoppings in the 1D momentum lattice (see Fig. 1b).

According to the mapping relation from the 1D momentum lattice to the ladder, the tunnelings in the $j = 0$ and $j = 1$ legs carry the phases of $\phi$ and $-\phi$, respectively, as shown in Fig. 1c. Hence, the phase accumulated by an atom following a closed-loop path around each triangular plaquette is $\phi$. By controlling these tunneling phases, we have full control over the synthetic flux.

### Mean-field interactions in momentum space
Originating from the short-range interatomic interactions, the interactions become long-ranged in the momentum space. Since quantum statistics leads to an exchange interaction for two colliding atoms in distinguishable momentum states[31,41], in the momentum lattice, the direct on-site interaction is characterized by $U/N$, while the interaction of two atoms on different sites is characterized by $2U/N$ (see Fig. 1c). The effective Hamiltonian describing the atomic interactions is

$$H_{\text{int}} = \frac{U}{2N} \sum_{\ell j} n_{\ell j}(n_{\ell j} - 1)$$
$$+ \frac{2U}{N} \left( \sum_{j' < j} \sum_{\ell, \ell' j} n_{\ell' j'} n_{\ell j} + \sum_{\ell' < \ell} \sum_{\ell j} n_{\ell' j} n_{\ell j} \right). \tag{6}$$

In writing Hamiltonian (6), a number of approximations are made. First, we neglect interaction processes that scatter atoms out of the discrete momentum modes resonantly coupled by the Bragg

transitions (altogether 14 discrete momentum states in the experiment). We then take a Hartree–Fock-like approximation in the momentum lattice, retaining only density–density contributions: the first term on the right-hand side of Eq. (6) is the direct on-site interaction; and the other two terms are interactions for atoms occupying different momentum–lattice sites. Here we neglect scattering processes involving more than two momentum–lattice sites, which are energetically suppressed compared to the density–density contributions, but may play a role through higher-order Bragg processes at longer evolution times. Importantly, since the number of atoms is much larger than that of the discrete momentum–lattice sites, the approximations above are reasonable, and widely adopted in the studies of the momentum lattice[31,32,35,40,41].

Now comparing the last two terms in Eq. (6), we see that a pair of atoms on distinguishable sites (momentum states) acquire additional exchange energy due to the boson statistics, in contrast to the case when both atoms are on the same site (momentum state). Physically, this is the origin of the effective on-site attractive interaction in the effective interaction Hamiltonian (2), which gives rise to self-trapping[31,41,44,45]. More explicitly, under the conservation of total atom number $N = \sum_{\ell,j} n_{\ell,j}$[31], Eq. (6) can be simplified to the interaction Hamiltonian (2).

For numerical simulations, we adopt a mean-field approach, setting $\langle c_{\ell,j} \rangle = \tilde{\varphi}_{\ell,j}$ in their Heisenberg equations of motion, and evolve the resulting coupled Gross–Pitaevskii equations to get the time-dependent condensate wavefunctions $\tilde{\varphi}_{\ell,j}$ (see Supplementary Note 3). The numerical results are in good agreement with the experimental measurements, thus confirming the approximations above, suggesting that the dominant interaction effect in the momentum space is well-captured by a density-dependent, local attractive potential, as dictated by the effective interaction Hamiltonian (2).

### Tuning the interactions

To study the interaction effect on the chiral dynamics of atoms in the synthetic ladder, the $s$-wave scattering length of $^{133}$Cs atoms is continuously tuned by a broad Feshbach resonance centered at the magnetic field of $B = -11.7$ G[7]. In each experimental cycle, the scattering length is quickly tuned to the target value before quenching the synthetic ladder. The strength of atomic interaction is characterized by the ratio of the mean-field interaction energy $U$ to the fixed inter-leg tunneling energy $J/h = 0.4$ kHz. Here, the mean-field interaction energy is given as $U = (4\pi\hbar^2 a/M)\rho$ with the scattering length $a$, the atomic mass $M$ and the effective atomic density $\rho$. In all experimental measurements, $\rho = 4 \times 10^{13}$ cm$^{-3}$ is used to determine $U$.

### Measuring density distribution in the synthetic ladder

All condensed atoms are first initialized in the site (0, 0) with zero momentum, and then are quenched to the synthetic ladder configuration under various interactions and fluxes. At the end of the time evolution, we switch off all laser fields and quickly tune the magnetic field to 17 G, where the zero-crossing scattering length is achieved. The momentum distribution of atoms is obtained by taking an optical density image after 22-ms time of flight. We extract the population of atoms in each site of the ladder according to the mapping relation from the 1D momentum lattice to the synthetic ladder.

## Data availability

The data that support the findings of this study are available from the corresponding authors upon request.

## Code availability

The codes that support the findings of this study are available from the corresponding authors.

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

## Acknowledgements

We acknowledge Feng Mei, Zhijian Li, and Jiazhong Hu for their discussions and comments on the paper. This work is supported by the Innovation Program for Quantum Science and Technology (Grant Nos. 2021ZD0302103 (J.M.), 2021ZD0301904 (W.Y.)), the National Natural Science Foundation of China (Grant Nos. 62020106014 (J.M.), 92165106 (Y.L.), 62175140 (Y.W.) and 11974331 (W.Y.)) and the Applied Basic Research Project of Shanxi Province (Grant No. 202203021224001 (Y.L.)).

## Author contributions

Y.L., H.D., Y.W., and J.M. contributed to the executions of the experiments. Y.L., Y.W., J.L., and W.Y. developed the theoretical model. L.X. and S.J. supervised the project. All authors discussed the results, contributed to the data analysis, and co-wrote the paper.

## Competing interests

The authors declare no competing interests.
