## [Peer Review File · Nature Communications]

Observation of frustrated chiral dynamics in an interacting triangular flux ladderREVIEWER COMMENTS

Reviewer #1 (Remarks to the Author):

In their manuscript, Li et al. describe an experiment in which they effectively realized a BEC in a frustrated triangular flux ladder, using the momentum-space mapping. The experimental protocol is elegant and interesting.

I would like to point out a major problem of the manuscript: It's related to the way the authors treat their interactions. In Eq. (1), the authors claim to have *point-like* local interactions among the bosons, proportional to the sum of squares of the local density. Is this really the Hamiltonian the authors believe they implemented? In Fig. 1 c, they describe non-local interactions — which indeed their experiment must realize! In the main text, the authors refer to the methods for a derivation of the effective interactions: They state that Eq. (1) can be obtained from Eq. (5), which I doubt. The notation in Eq. (5) is very confusing, what does it mean to have $l, j' < j$? Does that mean both l and j' should be smaller than j ? Or any l , but j' which must itself be smaller than j ? Moreover, I believe even Eq. (5) is wrong, interactions between different l and different j do not seem to appear in the sum. It further remains unclear why the microscopic point-like interactions underlying the experiment would not give rise to scattering-type non-linear processes, e.g. $a_{k+q}^{\dagger} a_{k'-q}^{\dagger} a_{k'} a_k$? The authors mention a 'mean-field' interaction, but the relevance of scattering-type terms between different k states (which map onto the ladder sites) is not discussed.

The use of the term 'dynamical phase diagram' is extremely sloppy: Phase diagrams should distinguish different phases separated by non-analyticities of some sort. In the present case, I don't think any symmetry is truly spontaneously broken dynamically. So the term 'phase diagram' should not be used.

The work is also closely related to earlier experiments with a quantum gas microscope, where interaction-induced chiral dynamics has been observed in 2-leg flux ladders, not on a mean-field level but in a system with exactly two bosons and truly point-like interactions [Tai et al., Nature 546, Pp. 519 (2017)]. The authors should cite this paper accordingly. Likewise, recent experiments exploring the interplay of strong interactions and synthetic gauge fields (e.g. [Clark et al., Nature 582, 41-45, (2020)] or [Leonard et al., <https://arxiv.org/abs/2210.10919>]) are not mentioned.

The authors mention that the interactions lead to self-trapping of the BEC, but it remains unclear how the non-local interactions (whose precise form I cannot understand, see above), lead to self-trapping.

In the caption of Fig. 1 it would be useful if the authors could describe the initial state used in their experiment.

In Fig. 3, it would be useful if the authors could clarify whether panels c, d and e are calculations or measurements.

Where the numerical calculations in Fig. 3 f, g, h performed for the correct interactions, or the wrong local interactions from Eq. (1)?

In summary, very significant scientific questions remain that need to be clarified. In the present form I cannot recommend publication of this work.

Reviewer #2 (Remarks to the Author):

The paper by Li et al. reports experimental results on chiral dynamics in flux ladders realized using cold atoms in laser-induced momentum lattices. The paper is well written and the results appear to be sound. Studies of chiral and frustrated dynamics using ultra-cold atoms have attracted considerable attention in recent years, so the results of the present paper will certainly be of interest to the community.

While I can generally recommend publication of the paper, I would like the authors to consider revising their description of the momentum lattice model, which is not very clear. and especially the method for mapping the momentum states into the virtual "sites". For instance, it would be helpful to see an example of some raw data, i.e. an actual CCD image and the way this is then turned into the lattice-site images shown in Fig. 1 d-f.

Also, in the last sentence of the discussion and outlook the authors mention that in future studies one might explore "flux lattices with inhomogeneous gauge fields or more exotic geometries." It would be nice if the authors could elaborate a bit more on this and outline how such lattices might be realized using their technique.

Reply to Reviewer 1

We thank Reviewer 1 for the constructive comments and suggestions. Below please find our point-by-point response.

In their manuscript, Li et al. describe an experiment in which they effectively realized a BEC in a frustrated triangular flux ladder, using the momentum-space mapping. The experimental protocol is elegant and interesting.

We thank the Reviewer for finding our experiment elegant and interesting, and for the constructive comments and suggestions.

I would like to point out a major problem of the manuscript: It's related to the way the authors treat their interactions. In Eq. (1), the authors claim to have *point-like* local interactions among the bosons, proportional to the sum of squares of the local density. Is this really the Hamiltonian the authors believe they implemented? In Fig. 1 c, they describe non-local interactions, which indeed their experiment must realize! In the main text, the authors refer to the methods for a derivation of the effective interactions: They state that Eq. (1) can be obtained from Eq. (5), which I doubt. The notation in Eq. (5) is very confusing, what does it mean to have $\ell, j' < j$? Does that mean both ℓ and j' should be smaller than j ? Or any ℓ , but j' which must itself be smaller than j ? Moreover, I believe even Eq. (5) is wrong, interactions between different ℓ *and* different j do not seem to appear in the sum. It further remains unclear why the microscopic point-like interactions underlying the experiment would not give rise to scattering-type non-linear processes, e.g. $a_{k+q}^+ a_{k'-q}^+ a_{k'} a_k$? The authors mention a 'mean-field' interaction, but the relevance of scattering-type terms between different k states (which map onto the ladder sites) is not discussed.

We thank the Reviewer for the insightful comments and questions, which offer us the opportunity to clarify the model and the approximations that we use.

First of all, the Reviewer is correct in pointing out the typo in Eq. (5), which should read

$$H_{\text{int}} = \frac{U}{2N} \sum_{\ell,j} n_{\ell,j} (n_{\ell,j} - 1) + \frac{2U}{N} \left(\sum_{j' < j} \sum_{\ell, \ell', j} n_{\ell', j'} n_{\ell, j} + \sum_{\ell' < \ell} \sum_{\ell, j} n_{\ell', j} n_{\ell, j} \right), \quad (\text{R1})$$

where the first term is the direct on-site interaction, and the last two terms are interactions for atoms occupying different momentum-lattice sites. We note that all the numerical simulations in

our manuscript are carried out using the coupled Gross-Pitaevskii (GP) equations (see the revised Supplementary Information), and are not affected by the typo. We apologize for the confusion.

Second, Eq. (R1) differs from the interaction terms in Hamiltonian (1) of the main text only by a constant energy shift. In fact, from Eq. (R1), we see that a pair of atoms in distinguishable sites (momentum states) acquire additional exchange energy due to the boson statistics, in contrast to the case when both atoms are on the same site (momentum state). Physically, this is the origin of the effective on-site attractive interaction in Hamiltonian (1).

More explicitly, we have

$$\begin{aligned} H_{\text{int}} &= \frac{U}{N} \sum_{\ell,j} n_{\ell,j} (n_{\ell,j} - 1)/2 + \frac{2U}{N} \left(\sum_{j' < j} \sum_{\ell, \ell', j} n_{\ell', j'} n_{\ell, j} + \sum_{\ell' < \ell} \sum_{\ell, j} n_{\ell', j} n_{\ell, j} \right) \\ &= U(N - \frac{1}{2}) - \frac{U}{2N} \sum_{\ell, j} n_{\ell, j}^2, \end{aligned} \quad (\text{R2})$$

where we have used the total-atom-number constraint $N = \sum_{\ell, j} n_{\ell, j}$. Hence, the interaction terms in Hamiltonian (1) differ from those in Eq. (R1) by a constant energy shift $U(N - \frac{1}{2})$, which does not affect the population dynamics of atoms in the synthetic ladder. Note that such a connection was pointed out in [Phys. Rev. Lett. 127, 130401 (2021)].

Furthermore, the effective interaction Hamiltonian of Eq. (R1) is the result of a number of approximations. As pointed out by the Reviewer, the inter-atomic interactions, dominated by the *s*-wave collisions, are point-like (known as the contact interactions) in the real space, which translate to non-local interactions in the momentum space. For atoms initialized in the zero-momentum state and subject to the Bragg transitions as in our experiment, we first neglect interaction processes that scatter atoms out of the discrete set of 14 momentum modes considered in our work. We then take a Hartree-Fock-like approximation in the resulting momentum lattice, retaining only density-density contributions [either on-site, as the first term of Eq. (R1); or off-site, as the remaining terms of Eq. (R1)]. The “scattering-type non-linear processes” that the Reviewer questioned are dropped at this stage (except for the terms with $q = 0$). In the absence of Bragg transitions, these scattering processes are energetically suppressed compared to the density-density contributions, but may play a role through higher-order Bragg processes, which may become important at longer evolution times when populations in multiple momentum states become appreciable. A systematic study of this effect is interesting but demanding (both experimentally and theoretically), and is beyond the scope of our current work.

Importantly, since the number of atoms is much larger than that of the discrete momentum-lattice sites, the approximations above are reasonable, and widely adopted in previous studies of

the momentum lattice [c.f. Phys. Rev. Lett. 120, 040407 (2018); Phys. Rev. X. 8, 031045 (2018); Phys. Rev. Lett. 124, 050502 (2020); Phys. Rev. Lett. 127, 130401 (2021); Phys. Rev. Lett. 129, 103401 (2022)], where the agreement between the experimental measurements and numerical simulations is good in general.

For numerical simulations, we adopt a mean-field approach, setting $\langle c_{\ell,j} \rangle = \tilde{\varphi}_{\ell,j}$ in the Heisenberg equations of motion of the field operators, and evolve the resulting GP equations to get the time-dependent condensate wavefunction $\tilde{\varphi}_{\ell,j}$ (see the revised Supplementary Information for details). The numerical results are in good agreement with the experimental measurements, thus confirming the approximations above, suggesting that the dominant interaction effect is well-captured by a density-dependent, local attractive potential in the GP equations, which are a direct result of Hamiltonian (1) under the mean-field approximation.

In the revised manuscript and Supplementary Information, we added discussions on the derivation of the coupled GP equations and the validity of the second-quantized interaction Hamiltonians. We also clarified the approximations that we made and the potential impact of the scattering terms mentioned by the Reviewer. We hope the Reviewer will be satisfied with these additions.

The use of the term ‘dynamical phase diagram’ is extremely sloppy: Phase diagrams should distinguish different phases separated by non-analyticities of some sort. In the present case, I don’t think any symmetry is truly spontaneously broken dynamically. So the term ‘phase diagram’ should not be used.

We understand the concerns of the Reviewer. In the revised manuscript, we no longer use the term “dynamic phase diagram”. Instead, we now claim to have systematically characterized the dynamic behavior, revealing typical dynamic patterns of the system under different parameters.

The work is also closely related to earlier experiments with a quantum gas microscope, where interaction-induced chiral dynamics has been observed in 2-leg flux ladders, not on a mean-field level but in a system with exactly two bosons and truly point-like interactions [Tai et al., Nature 546, Pp. 519 (2017)]. The authors should cite this paper accordingly. Likewise, recent experiments exploring the interplay of strong interactions and synthetic gauge fields (e.g. [Clark et al., Nature 582, 41-45, (2020)] or [Leonard et al., <https://arxiv.org/abs/2210.10919>]) are not mentioned.

We thank Reviewer 1 for reminding us of these important works, which we cited and discussed in the revised Introduction.

The authors mention that the interactions lead to self-trapping of the BEC, but it remains unclear how the non-local interactions (whose precise form I cannot understand, see above), lead to self-trapping.

Following the derivation and discussion in our response above, the impact of the non-local interaction in the momentum space can be approximated by an on-site attractive potential. The underlying mechanism is the reduction in the exchange energy when interacting atoms occupy the same momentum state. It follows that, for atoms initialized in a given momentum state [the site ($\ell = 0, j = 0$) in our experiment], they effectively experience a density-dependent, local attractive potential. As such, when the on-site density is sufficiently large and/or the interaction strength is sufficiently strong, the hopping processes are suppressed and the atoms localize at the initial site, as predicted by self trapping.

In the caption of Fig. 1 it would be useful if the authors could describe the initial state used in their experiment.

We thank the Reviewer for the helpful suggestion. In the revised manuscript, we have added the following description in the caption of Fig. 1: “For all experiments and numerical simulations, we initialize the atoms on the site ($\ell = 0, j = 0$), corresponding to the ground state of a non-interacting BEC.”

In Fig. 3, it would be useful if the authors could clarify whether panels c, d and e are calculations or measurements.

Panels c, d, and e in Fig. 3 are from experimental measurements. We have clarified this in the figure caption.

Where the numerical calculations in Fig. 3 f, g, h performed for the correct interactions, or the wrong local interactions from Eq. (1)?

All numerical simulations presented in this work are carried out using the coupled GP equations [Eq. (S9) in the revised Supplementary Information] that can be derived from Hamiltonian (S10) whose interaction terms are the same as those in Eq. (5) in the Methods. As we discussed in our response above, Hamiltonian (S10) is different from Hamiltonian (1) only by a constant energy shift, so that both predict the same population dynamics of atoms in the synthetic ladder.

In the revised Supplementary Information, we have provided detailed derivations for the coupled GP equations that we use for numerical simulations. Note that similar GP equations and the effective second-quantized Hamiltonians have been widely used for momentum lattices [c.f. Phys. Rev. Lett. 120, 040407 (2018); Phys. Rev. Lett. 127, 130401 (2021); Phys. Rev. Lett. 129, 103401 (2022)], and were found to provide satisfactory descriptions for experimental measurements.

In summary, very significant scientific questions remain that need to be clarified. In the present form I cannot recommend publication of this work.

We thank the Reviewer for the constructive comments which give us the opportunity to further improve our manuscript. We hope the Reviewer will be satisfied with these changes.

Reply to Reviewer 2

We thank Reviewer 2 for the constructive comments and suggestions. In the following, let us address the Reviewer’s comments point-by-point.

The paper by Li et al. reports experimental results on chiral dynamics in flux ladders realized using cold atoms in laser-induced momentum lattices. The paper is well written and the results appear to be sound. Studies of chiral and frustrated dynamics using ultra-cold atoms have attracted considerable attention in recent years, so the results of the present paper will certainly be of interest to the community.

We thank Reviewer 2 for finding our work well-written and interesting.

While I can generally recommend publication of the paper, I would like the authors to consider revising their description of the momentum lattice model, which is not very clear. and especially the method for mapping the momentum states into the virtual “sites”. For instance, it would be helpful to see an example of some raw data, i.e. an actual CCD image and the way this is then turned into the lattice-site images shown in Fig. 1 d-f.

We appreciate the Reviewer’s recommendation and constructive comments.

In the revised manuscript, we explicitly discussed the mapping relation between the discrete momentum states, and sites in the triangular ladder. Specifically, in the left column of page 2, we have added “The mapping relation between the discrete momentum states and sites in the synthetic triangular ladder is given by $m = 2\ell + j$.” Further, following the suggestion of the Reviewer, we have added a figure (the new Fig. S1) in the Supplementary Information, showing the experimentally measured one-dimensional momentum distribution, as well as the density distribution in the synthetic ladder following the mapping.

Also, in the last sentence of the discussion and outlook the authors mention that in future studies one might explore “flux lattices with inhomogeneous gauge fields or more exotic geometries.” It would be nice if the authors could elaborate a bit more on this and outline how such lattices might be realized using their technique.

We thank the Reviewer for the helpful suggestion. In the revised Discussion section, we have added discussions on how to implement inhomogeneous gauge fields, as well as more exotic geometries.

Specifically, the newly added discussions read “Here the inhomogeneous gauge fields can be implemented by individually controlling the tunneling phase of each laser-assisted hopping process along the legs. More complicated lattice geometries can be achieved by parametrically coupling the desired momentum states with multi-photon processes, or by involving atomic hyperfine states into the lattice design.”

List of changes

1. We have added detailed derivations of the coupled Gross-Pitaevskii (GP) equations and the relevant second-quantized interaction Hamiltonians in the Supplementary Information. We have also added discussions on the approximations that we made.
2. We have replaced “dynamic phase diagram” with “dynamic behavior” or “typical dynamic pattern” throughout the manuscript.
3. Following the Reviewer 1’s suggestion, we added more references (see Refs. 10-12), along with relevant discussions in the Introduction.
4. We have revised the figure captions to address the Reviewers’ concerns.
5. We have added the mapping relation between the discrete momentum states and sites in the synthetic ladder on page 2, and showed an example in Fig. S1 using experimental data.
6. We have added discussions on the realization of inhomogeneous gauge fields and exotic geometries in the Discussion section.
7. In the revised manuscript (including the Supplementary Information), we have marked all major revisions in blue.

Reviewers' comments:

Reviewer #1 (Remarks to the Author):

I have read the referee reply by the authors. They have fixed a mistake in their Hamiltonian presented in the supplements, and provided a more detailed explanation of their model. While I now understand the connection between Eq. (1) and Eq. (5), I don't understand the underlying approximations, specifically why scattering terms such as $a^{\{k+q\}} + a_{\{k'-q\}} + a_{\{k'\}} a_k$ are neglected. The authors claim that such terms are 'energetically suppressed', but by what terms? Hamiltonian (1) only contains the bare kinetic energy (hopping between momenta k) and the interactions. When the interactions are strong, I don't see any further term that could energetically suppress the scattering terms I already mentioned in my first referee report.

The authors further explain that they make a Hartree-Fock type approximation in order to justify why they retain only density-density interaction terms. I understand what's done, but I don't understand why that's necessary: After all, the authors work with an effective Gross-Pitaevskii equation (GPE) for the mean-field amplitudes. On a mean-field level, it should be easy to include off-diagonal scattering terms like the ones mentioned above: Once they're normal-ordered, they can easily be turned into couplings in the GPE.

In their reply the authors claim that the good agreement of their numerics with experiment justifies their approximations. I find this reasoning a bit dangerous, although I do agree with the authors that the reported agreement is good, and it is nice to see that their model is able to qualitatively explain their observations.

More broadly, I am concerned about the claim of the authors that their system provides a quantum simulation of the Hamiltonian in Eq. (1), formulated in 2nd quantization. As discussed above / by the authors, their Eq. (1) really can only be justified on a mean-field level. Hence it is very misleading to claim that their experiment actually simulates this second-quantized Hamiltonian. The reason this is a contentious issue, is that other quantum simulation experiments with ultracold atoms in optical lattices actually do faithfully simulate the 2nd quantized Hamiltonian mentioned by the authors. Therefore, I believe the authors must formulate their effective model on a mean-field level, unless they can make a claim that beyond mean-field effects (such as governed by a 2nd quantized Hamiltonian) can be directly observed experimentally.

I am now convinced that the principle results and theoretical analysis performed by the authors is sound, and the paper can be published. However, I am not fully convinced that the obtained results are sufficiently impactful to warrant publication in a journal like Nature Communications. I have seen experimental papers reporting significantly more impressive results rejected from Nature Communications (where I was not personally involved as a co-author). The quantum simulation of the interplay of interactions and chiral dynamics is certainly a worthy goal, but I doubt that the mean-field

level analysis performed here raises to the level of importance that would justify publication in Nature Communications.

Reviewer #2 (Remarks to the Author):

In the revised version, the authors have addressed all of points I raised in my first review. Specifically, they have described the mapping relation for the momentum states more clearly and have also added a more detailed discussion section. I can now recommend the paper for publication.

Response to the review reports of the manuscript NCOMMS-23-14903-A
“Observation of frustrated chiral dynamics in an interacting triangular flux ladder”

List of changes

1. We have revised the theoretical-model section of the main text (marked by using red fonts), to address Reviewer 1’s comments.

Reply to Reviewer 1

We thank Reviewer 1 for his/her time. It is regrettable that the Reviewer was not fully convinced to recommend our work for publication. While the Reviewer's comments focus on the various technical issues regarding the mean-field analysis, we wish to emphasize that our work is above all experimental. The experiment, by design, involves the interplay of synthetic flux and interaction in a frustrated geometry, and our observation, while necessarily affected by the beyond-mean-field effects of interaction, clearly demonstrates the impact of interaction and frustrated geometry on the chiral dynamics. Such an observation is the central finding of our experiment, which, being the first of its kind, would surely stimulate further studies on the subject.

On the other hand, to provide a clear (albeit simplified) physical picture to the observed phenomena, we adopt a mean-field description, which is also adopted in all existing experimental literature on the momentum lattice [see for instance, Refs. 31,32,35,40,42,43]. The numerical simulations following such a treatment are in qualitative agreement with the experimental observation, suggesting: i) the dominant interaction effect in the momentum space is well captured by a mean-field approximation; ii) a more elaborate, beyond-mean-field theoretical description is needed for quantitative explanation of our observation. While the former firmly establishes the overall physical picture on the interplay of interaction and chiral dynamics (which is the key message of our experiment), the latter further highlights the necessity of developing beyond-mean-field models, now that the momentum-lattice engineering is enriched by tunable interactions through the Feshbach resonance (as pioneered in our experimental system). This would definitely stimulate future theoretical studies.

As such, we believe our experiment constitutes the state-of-the-art of momentum-lattice engineering, and has significant impact on the quantum simulation of correlated transport in frustrated geometries.

I have read the referee reply by the authors. They have fixed a mistake in their Hamiltonian presented in the supplements, and provided a more detailed explanation of their model. While I now understand the connection between Eq. (1) and Eq. (5), I don't understand the underlying approximations, specifically why scattering terms such as $a_{k+q}^\dagger a_{k'-q}^\dagger a_{k'} a_k$ are neglected. The authors claim that such terms are 'energetically suppressed', but by what terms? Hamiltonian (1) only contains the bare kinetic energy (hopping between momenta k) and the interactions. When the interactions are strong, I don't see any further term that could energetically suppress

the scattering terms I already mentioned in my first referee report. The authors further explain that they make a Hartree-Fock type approximation in order to justify why they retain only density-density interaction terms. I understand what's done, but I don't understand why that's necessary: After all, the authors work with an effective Gross-Pitaevskii equation (GPE) for the mean-field amplitudes. On a mean-field level, it should be easy to include off-diagonal scattering terms like the ones mentioned above: Once they're normal-ordered, they can easily be turned into couplings in the GPE. In their reply the authors claim that the good agreement of their numerics with experiment justifies their approximations. I find this reasoning a bit dangerous, although I do agree with the authors that the reported agreement is good, and it is nice to see that their model is able to qualitatively explain their observations.

In the absence of the Bragg coupling fields, the scattering terms such as $a_{k+q}^\dagger a_{k'-q}^\dagger a_{k'} a_k$ are offset by the kinetic energy difference $\frac{\hbar^2}{2M}[(k+q)^2 + (k'-q)^2 - k'^2 - k^2]$. These processes are therefore suppressed compared to those that we consider, such as $a_k^\dagger a_{k'}^\dagger a_{k'} a_k$ (mode-preserving processes), whose incoming and outgoing scattering states have the same kinetic energy.

We emphasize that such treatments are the current state-of-the-art in the modeling of momentum-lattice interaction effects, and are adopted to analyze a wide range of seminal experiments [see for instance, Phys. Rev. Lett. 120, 040407 (2018), Phys. Rev. X 8, 031045 (2018), Phys. Rev. Lett. 124, 050502 (2020), Phys. Rev. Lett. 124, 070402 (2020), Phys. Rev. Lett. 126, 040603 (2021), Phys. Rev. Lett. 127, 130401 (2021), Phys. Rev. Lett. 129, 103401 (2022)]. Similar to the practice in the literature, we adopt the mean-field description, not to offer a quantitative fit of the data, but to provide a qualitatively correct and more transparent physical picture. The fact that the numerical results agree well (on a qualitative level) with the experimental observation suggests that quantum fluctuations (which necessarily exist in our experiments) do not change the overall physical picture, though a more elaborate theory is needed for quantitative explanations.

More broadly, I am concerned about the claim of the authors that their system provides a quantum simulation of the Hamiltonian in Eq. (1), formulated in 2nd quantization. As discussed above / by the authors, their Eq. (1) really can only be justified on a mean-field level. Hence it is very misleading to claim that their experiment actually simulates this second-quantized Hamiltonian. The reason this is a contentious issue, is that other quantum simulation experiments with ultracold atoms in optical lattices actually do faithfully simulate the 2nd quantized Hamiltonian mentioned by

the authors. Therefore, I believe the authors must formulate their effective model on a mean-field level, unless they can make a claim that beyond mean-field effects (such as governed by a 2nd quantized Hamiltonian) can be directly observed experimentally.

We are not claiming that our experiment provides a quantum simulation of the Hamiltonian in Eq. (1) [Eqs. (1)(2) in the revised manuscript]. Rather, our experiment directly simulates a scenario where the interplay of synthetic flux and tunable interaction in a frustrated geometry gives rise to novel dynamic patterns. Since the experiment is quantum mechanical in nature, the observed results necessarily include the impact of all possible scattering processes. We then adopt the approximate model Hamiltonian Eqs. (1)(2) and the mean-field approach to provide a qualitative understanding of the experimental system.

In the revised manuscript, we explicitly clarify the relation between experimental observation and our approximate model, as well as on the mean-field nature of our numerical simulations.

I am now convinced that the principle results and theoretical analysis performed by the authors is sound, and the paper can be published. However, I am not fully convinced that the obtained results are sufficiently impactful to warrant publication in a journal like Nature Communications. I have seen experimental papers reporting significantly more impressive results rejected from Nature Communications (where I was not personally involved as a co-author). The quantum simulation of the interplay of interactions and chiral dynamics is certainly a worthy goal, but I doubt that the mean-field level analysis performed here raises to the level of importance that would justify publication in Nature Communications.

We thank the reviewer for finding our result and analysis sound. We emphasize again, that since our work is experimental, the observed interplay of interaction and chiral dynamics itself is the key finding here, rather than our mean-field analysis. Such a mean-field analysis appears in all existing experimental studies of synthetic momentum lattice, and is not a novelty of our work. But we firmly believe our experimental characterization of the interplay of interaction and chiral dynamics is a breakthrough in the quantum simulation with synthetic momentum lattices, and thus suitable for publication in Nature Communications.

Reply to Reviewer 2

We thank Reviewer 2 for recommending our work for publication.

REVIEWERS' COMMENTS

Reviewer #1 (Remarks to the Author):

I have read the rebuttal by the authors. I agree with the authors that the effective Hamiltonian they use is routinely used in the literature they cited in their rebuttal, and I also understand now that most of the interaction terms scattering to different momentum modes can be neglected (the authors just make an argument in the case without Bragg couplings, which I first thought is confusing since they have Bragg couplings; but they are correct that this argument still applies with the Bragg couplings on, as I suppose one can see by going to a rotating frame of reference.) This brings me back to my main conclusion from the last round of review: As I concluded previously: "I am now convinced that the principle results and theoretical analysis performed by the authors is sound, and the paper can be published."

So it's really an editorial decision about relevance of the results that needs to be taken. I don't share the author's opinion that they achieved "a breakthrough in the quantum simulation with synthetic momentum lattices". For example, An, Meier and Gadway in [PRX 8, 031045 (2018)] have used the same technique to realize a synthetic flux ladder in a BEC dressed by light, and they too have observed chiral dynamics in this system. I'm not saying the present study is a copy of the previous study, but I simply fail to see a major breakthrough here, or how this study enables a major breakthrough in the near future. I emphasize that this work is addressing an interesting question which will be of interest to an expert audience, and I truly appreciate the experimental effort by the authors and believe this work deserves publication in a well-regarded but more specialized journal.

Reply to Reviewer 1

We thank Reviewer 1 for his/her time. In the following, we address the Reviewer's comment point-by-point.

I have read the rebuttal by the authors. I agree with the authors that the effective Hamiltonian they use is routinely used in the literature they cited in their rebuttal, and I also understand now that most of the interaction terms scattering to different momentum modes can be neglected (the authors just make an argument in the case without Bragg couplings, which I first thought is confusing since they have Bragg couplings; but they are correct that this argument still applies with the Bragg couplings on, as I suppose one can see by going to a rotating frame of reference.) This brings me back to my main conclusion from the last round of review: As I concluded previously: "I am now convinced that the principle results and theoretical analysis performed by the authors is sound, and the paper can be published." So it's really an editorial decision about relevance of the results that needs to be taken.

We are pleased that Reviewer 1 is satisfied with our response over his/her previous technical concerns.

I don't share the author's opinion that they achieved "a breakthrough in the quantum simulation with synthetic momentum lattices". For example, An, Meier and Gadway in [PRX 8, 031045 (2018)] have used the same technique to realize a synthetic flux ladder in a BEC dressed by light, and they too have observed chiral dynamics in this system. I'm not saying the present study is a copy of the previous study, but I simply fail to see a major breakthrough here, or how this study enables a major breakthrough in the near future. I emphasize that this work is addressing an interesting question which will be of interest to an expert audience, and I truly appreciate the experimental effort by the authors and believe this work deserves publication in a well-regarded but more specialized journal.

We appreciate the Reviewer finding our work interesting and publishable in well-regarded journals. As we have pointed out in the Introduction, while chiral dynamics in a synthetic flux ladder was reported in previous studies such as [PRX 8, 031045 (2018), cited as Ref. 35 in our work], the interplay of interaction, geometric frustration and synthetic flux was largely unexplored, since

the interactions therein are weak and with little tunable range. We believe that our work pioneers in the combination of widely tunable interaction and frustrated geometry through momentum-lattice engineering. Such a combination not only paves the way for systematic studies of correlated transport in exotic geometries, but also foreshadows the necessity of developing beyond-mean-field descriptions which would be indispensable for simulating and understanding more complex correlated transport in momentum lattices. As such, we trust our work represents a significant advance, and would stimulate further theoretical and experimental studies in various communities including cold atoms, quantum simulation and condensed matter.

At the end of the Discussion section, we have added a sentence to emphasize the potential impact of the experiment. We believe this new iteration is suitable for the wide audience of *Nature Communications*.